# An Analysis of the Colony Structure of Prokaryotes in the Jialing River Waters in Chongqing

**DOI:** 10.3390/ijerph19095525

**Published:** 2022-05-02

**Authors:** Maolan Zhang, Guoming Zeng, Dong Liang, Yiran Xu, Yan Li, Xin Huang, Yonggang Ma, Fei Wang, Chenhui Liao, Cheng Tang, Hong Li, Yunzhu Pan, Da Sun

**Affiliations:** 1Chongqing Engineering Laboratory of Nano/Micro Biological Medicine Detection Technology, School of Architecture and Engineering, Chongqing University of Science and Technology, Chongqing 401331, China; 2019016@cqust.edu.cn (M.Z.); 2017015@cqust.edu.cn (G.Z.); l632413306@163.com (D.L.); x8532105@163.com (Y.X.); 2019443106@cqust.edu.cn (X.H.); 2021206106@cqust.edu.cn (F.W.); 2021206103@cqust.edu.cn (C.L.); 2021206010@cqust.edu.cn (C.T.); 2School of Pharmacy, Taizhou Polytechnic College, Taizhou 225300, China; 20066954@cqu.edu.cn (Y.L.); zml@cqu.edu.cn (Y.M.); 3Key Laboratory of Eco-Environment of Three Gorges Region, Ministry of Education, Chongqing University, Chongqing 400044, China; 4School of Chemical Engineering, Sichuan University of Science and Engineering, Zigong 643000, China; 5Institute of Life Sciences and Biomedical Collaborative Innovation Center of Zhejiang Province, Wenzhou University, Wenzhou 325035, China

**Keywords:** prokaryotes, high throughput sequencing, biological information analysis, community structure

## Abstract

At present, research on the influence of human activities (especially urbanization) on the microbial diversity, structural composition, and spatial distribution of rivers is limited. In this paper, to explore the prokaryotic community structure and the relationship between the community and environmental factors in the Jialing River Basin of Chongqing, so as to provide a basis for monitoring microorganisms in the watershed. The V3–V4 region of the 16 S rRNA gene was analyzed by high-throughput sequencing and the microbial community of the waters of the Jialing River was analyzed for the diversity and composition of the prokaryotic community as well as the species difference of four samples and correlations with environmental factors. The main results of this study were as follows: (1) The diversity index showed that there were significant differences in the biodiversity among the four regions. At the genus level, *Limnohabitans*, *unclassified_f_Comamonadaceae*, and *Hgcl_clade* were the main dominant flora with a high abundance and evenness. (2) A Kruskal–Wallis H test was used to analyze the differences of species composition among the communities and the following conclusions were drawn: each group contained a relatively high abundance of *Limnohabitans*; the Shapingba District had a higher abundance of *Limnohabitans*, the Hechuan District had a wide range of *unclassified_f_Comamonadaceae*, and the Beibei District had a higher *Hgcl_clade*. (3) Through the determination of the physical and chemical indicators of the water—namely, total nitrogen, total phosphorus, chemical oxygen demand, chlorophyll A, and an analysis by an RDA diagram, the results demonstrated that the distribution of microbial colonies was significantly affected by the environmental factors of the water. Chemical oxygen demand and ammonia nitrogen had a significant influence on the distribution of the colonies. Different biological colonies were also affected by different environmental factors.

## 1. Introduction

Microbial communities in aquatic ecosystems play an important role in degrading organic matter and participating in nutrient cycling, but they are sensitive to the external environment [1,2]. At present, the research on bacterial community diversity and its relationship with environmental factors is growing [3]. However, most studies focus on coastal areas, lakes, and oceans [4,5], and there are few studies on rivers. Due to the increasingly serious pollution of urban rivers, the response of aquatic biota in urban rivers has gradually attracted researcher’s attention.

Rivers are an important part of the global water cycle and one of the important water sources for human industrial and agricultural production as well as for life; they can provide an early warning to the overall quality of the water environment [6]. The Jialing River Basin in Chongqing is an ecological security barrier in the southwest of the country. The Jialing River is an important water source for industrial and agricultural production as well as domestic water and fishery in the urban area of Chongqing. It is also an important water source conservation area in the upper reaches of the Yangtze River. The water quality of the Jialing River affects the economic and social development of Chongqing [7]. Therefore, it is significant to study the characteristics of flora in the Jialing River Basin.

Microorganisms are an important part of ecosystems and are highly adaptable and prone to variation. Their structures and functions follow the changes of environmental factors. Microorganisms can maintain the overall stability of the ecosystem [8,9] and play a vital role in water quality control [10]. They have significant ecological functions in nutrient cycling, pollution reduction, soil protection, climate regulation, and water conservation [11]. As a key part of the composition of the biosphere, bacteria play an extraordinary role in the flow of matter and energy. The compositional diversity, distribution characteristics and inter-community structure of bacterial communities in lake ecosystems are the basis and prerequisite for understanding the structure and maintenance mechanism of lake ecosystems, and provide an important basis for basic theoretical research and environmental monitoring of lake microbial ecology.

In this paper, the research method based on high-throughput sequencing was used to conduct sampling at four sampling points in the Jialing River Basin, the Hechuan District, the Beibei District, the Shapingba District, and the Jiangbei District. The total microbial DNA was extracted from the samples, a PCR library was constructed, and the V3–V4 of the fragment 16S rDNA fragment was used as the research object. The structural diversity of bacterial flora in Jialing River water was analyzed by high-throughput sequencing, their relationship with environmental factors was analyzed by redundancy analysis (RDA) [12], and the biological community was analyzed by bioinformatics and metagenomics methods. To explore the prokaryotic community structure and the relationship between the community and environmental factors in the upper, middle, and lower reaches of the Jialing River Basin, so as to provide a theoretical basis for microbial monitoring in the entire river basin in Chongqing.

## 2. Materials and Methods

### 2.1. Experimental Materials

A bacterial genomic DNA extraction kit (Baimaike Biotechnology Co., Ltd., Beijing, China) and a PCR kit (Baimaike Biotechnology Co., Ltd., Beijing, China) were used in addition to sulfuric acid, nitric acid, potassium persulfate, phenolphthalein, potassium dihydrogen phosphate, sodium hydroxide, potassium iodide, potassium dichromate, potassium dihydrogen phosphate, and acetone (Meiji Biotechnology Co., Ltd., Shanghai, China). All chemical reagents of absolute ethanol, boric acid, hydrochloric acid, and potassium phthalate (Meiji Biotechnology Co., Ltd., Shanghai, China) are of analytical grade. A vertical pressure steam sterilizer (YXQ-LS-50A), UV-Vis spectrophotometer (TU-1901, T68), and small Benchtop Refrigerated Centrifuge (Centrifμge 5418R) were also used in the experiments.

### 2.2. Sample Collection

The sample collection took place between July–September 2020, as this was the high-water season of the Jialing River. To analyze the characteristics of microbial colonies in the Chongqing area of the Jialing River, four sampling points (as shown in Figure 1) were selected for sampling. The four sampling points were the Nanping Bridge of the Jialing River in the Hechuan District (upstream, indicated as H), the Beibei Jialing River Bridge (upper and middle reaches, indicated as B), the Shapingba Shimen Bridge (middle and lower reaches, indicated as S), and the Jiangbei QianSimen Bridge (downstream, indicated as J). Each sampling point is divided into upper, middle, and lower vertical levels. Due to the large variation in the river water depth, the upper sampling depth (height from the river water surface) was approximately 0.5 m, the middle layer was approximately 3–5 m, and the lower layer was approximately 5–10 m. The tertiary water was then mixed and repeated three times for each sampling point. A total of 1 L of the water sample was filtered through a 0.22 μm microporous membrane and stored at −80 °C. This was then sent to a sequencing company for 16S rDNA transcriptome sequencing.

### 2.3. Determination of Physical and Chemical Indicators of Water Bodies

Four sampling points were in the Jialing River Basin, the Hechuan District, the Beibei District, the Shapingba District, and the Jiangbei District. The reason for choosing Shapingba District and Jiangbei District is that they belong to the upper end of the Jialing River and have a large population base. Beibei District belongs to the middle section of the Jiangling River and has a medium population. Hechuan District belongs to the lower end of the Jialing River, which is a suburb and has a small population. The determination of physical and chemical indicators of the samples obtained, water quality total nitrogen (GB-11894-1989), total phosphorus (GB-11893-1989), ammonia nitrogen (GB-7479-87), chemical oxygen demand (GB-11914-1989), and chlorophyll A (HJ-897-2017) were performed in accordance with the national standards.

### 2.4. Bacterial Genomic DNA Extraction

The total bacterial DNA extraction was performed according to the Bacterial Genomic DNA Extraction Kit Manual in Table 1. Electrophoresis of 1% agarose gel was then used to detect the amount of DNA extracted, and NanoDrop2000 (micro-ultraviolet spectrophotometer) was used to determine the concentration and purity of DNA. A total of 100 μL of buffer B1 was added to 0.5 mL of a bacterial solution, and then 100 μL of buffer B2 were added. This mixture was well-shaken, placed in a centrifuge, and centrifuged at 12,000 rpm/min for 2 min. After centrifugation, 100 μL of the supernatant were pipetted into another clean centrifuge tube for use as a template.

### 2.5. PCR Amplification

Bacterial 16S rDNA PCR operation steps, establishment of a PCR reaction system, and 20 μL system in Table 2. After adding TransStart FastPfu buffer, dNTPs, upstream primers, downstream primers, TransStart FastPfu DNA polymerase, and template DNA were added to produce a 20 μL mixture. All reagents were collected at the bottom of the tube by brief centrifugation. The 16S rRNA gene V3-V4 variable region was PCR amplified using 338F(5′-ACTCCTACGGGAGGCAGCAG-3′) and 806R (5′-GGACTACHVGGGTWTCTAAT-3′) in triplicate per sample.

### 2.6. Data Statistical Analysis

The bases were filtered with a tail quality value below 20 of the reads, a 50 bp window was set, and the reads were filtered below 50 bp after quality control. The reads containing N bases were then removed, according to the overlap relationship between the PE reads. The paired reads were then merged into one sequence; the minimum overlap length was 10 bp, and the maximum mismatch ratio allowed in the overlap region of the spliced sequence was 0.2. The samples were distinguished according to the barcodes and primers at the beginning and end of the sequence, and the sequence direction was adjusted. The number of mismatches allowed by a barcode was 0, and the maximum number of primer mismatches was 2. Using UPARSE software (version 7.1) [13], the sequences were OTU clustered with a similarity of 97% [14,15], and chimeras were eliminated. The RDP classifier (version 2.2) [13] was used to classify and annotate each sequence, and then compared with the Silva 16S rRNA database (v138) with an alignment threshold of 70%. Solexa was then used, which is sequencing-by-synthesis (SBS) technology that uses DNA clustering, bridge PCR, and reversible blocking methods to read the base sequences in DNA in turn. Optical equipment is used to read and record the fluorescent signal; finally, the computer converts the analysis results into sequencing information.

The data obtained in the experiment were analyzed and processed using the Origin 8.0 software, OriginLab, (OriginLab, Redmond, WA, USA) processing system, and Windows Excel and Word (2003, 2010 editions) office software, (Microsoft, OriginLab, Redmond, WA, USA.) All determinations were carried out in triplicate, and the mean values were presented. The data are reported as the averages of three separate experiments ± SD (*n* = 3).

## 3. Results and Discussion

### 3.1. Water Quality Measurement Results

As important freshwater habitats for microorganisms, urban rivers are mainly affected by the terrestrial environment and human activities. When pollutants flow into the river, basic parameters such as pH, temperature, dissolved oxygen, salinity, and nitrogen and phosphorus content in the water body change, which is of great significance for regulating the complex microbial community and further reshapes the bacterial community structure [16,17,18]. For the microbial community structure at the water–sediment interface found in the Chaka Salt Lake (China) study, the apparent changes in the lake water salinity were associated with a decrease in lake water [19]; studies in the Ebro River (Iberian Peninsula, Spain) revealed that electrical conductivity, temperature, and dissolved inorganic nitrogen are the main environmental factors affecting the composition of phytoplankton communities [20]. The water quality indicators of the sampled water bodies were measured; the results are shown in Table 3. The sampling areas had obvious differences, particularly the Hechuan District, which had a high ammonia nitrogen content, a high total nitrogen content, and a low chemical oxygen demand. Furthermore, by monitoring the water quality of the four sampling points, it was found that the water quality of the four points was polluted to a certain extent. The indicators of nitrogen and phosphorus exceeded the normal values of the water body; in particular, the indicators of algae and chlorophyll a exceeded the average value of the river body. Therefore, it is speculated that the four sampling points are prone to the risk of water eutrophication in summer and autumn. Changes in the river environment can affect the balance between alien bacteria introduced by surface runoff and indigenous communities, predation or competition in the food chain, and directly or indirectly affect the function of bacterial communities [21]. Therefore, understanding river bacterial community diversity and spatial distribution has profound implications for monitoring ecosystem health.

### 3.2. Alpha/Beat Diversity Analysis

Alpha analysis is a composition analysis—the similarity of the species composition of each sample or group could be reflected. Determining bacterial metabolic potential through river bacterial diversity is an extremely important but under-recognized scientific issue. Changes in the bacterial structure of rivers can further lead to changes in metabolic and environmental information processing pathways [22,23]. From the upper to the lower reaches of the Danube, the relative abundance of its typical freshwater flora gradually increased due to a decrease in microbial richness and evenness [24]. The microbial diversity of the Yenisei peaked in the middle section, which may be associated with high flow and rapid turbulence [25]. As seen in Figure 2, the Shannon indices of the samples were all between 3.2 and 4.5 and the species composition richness and diversity of the four groups were relatively high. PCA analysis showed that the four groups could clearly be distinguished, indicating that the four groups had significant differences in the composition of the biological community. The R value was relatively close to 1, indicating that the difference between the groups was slightly greater than the difference within the group. There were obvious differences in the community composition structure of the four groups and the community composition in the different regions was different. The reason for this may be that Jiangbei District has the largest industrial area, agricultural area, and population, which has a greater impact on water bodies, so the diversity of bacterial species is the highest; Beibei District and Hechuan District are rich in ecological communities, so the microorganisms in the riparian zone may be due to rainwater washed into the lake, so the microbial community is relatively rich, while the Shapingba area has the weakest community diversity because the industrial area, agricultural area, and ecological community zone are relatively less than the above-mentioned sampling points. Therefore, identifying the bacterial diversity of rivers can help to determine the impact of the metabolic potential of river bacterial communities.

### 3.3. Community Composition Analysis

#### 3.3.1. Venn Diagram Analysis

A Venn diagram permits a macroscopic reflection of the number of species in the community through an intersection, which can reflect the composition of species. As seen in Figure 3, the Hechuan District and Beibei District shared 308 species, the Hechuan District and Shapingba District shared 217 species, the Hechuan District and Jiangbei District trict shared 319 species, the Beibei District and Shapingba District shared 243 species, the Beibei District and Jiangbei District shared 232 species, and the Shapingba District and Jiangbei District share 248 species. The Beibei District had the greatest number of distinct species (109 species), and the Shapingba District had the smallest number of distinct species (9 species). This may be due to riparian influences leading to the proliferation of exotic bacteria (i.e., soil and groundwater) and indigenous flora (i.e., increased bacterial richness). For example, studies on the Mississippi River (US) [26], Thames River (UK) [27], and Amazon River [28] have found that the “heterologous input process” or “mass effect” in the riparian zone further indicates that the diffusion rate of organic matter exceeds the local bacterial extinction rate, making the exogenous input bacteria [29,30,31]. The diversity is much higher than the river itself.

#### 3.3.2. Community Bar Chart Analysis

Through a bar chart analysis, it could be seen that the measured water samples had a significantly high bacterial diversity. Figure 4 showed the bar map at the phylum level. The community composition structure of the Beibei District and Jiangbei District was relatively similar. The higher relative abundance in the Hechuan District were *unclassified**-f**-Comamonadaceae* [32], *unclassified**-f**-Rhodobacteraceae* [33], and the *Hgcl_clade* [34]. The dominant flora was the *Hgcl_clade*, and its abundance in the three samples in this group was 12.35, 12.25, and 15.28%, respectively. The relatively high relative abundance levels in the Beibei District were *unclassified_f_Comamonadaceae*, the *Hgcl**_clade*, and *Acinetobacter*. The dominant flora was *Hgcl-clade*, whose abundances in the three samples were 12.98, 10.78, and 11.57%, respectively. In the Shapingba District, the relative abundances form *Limnohabitans*, *Acinetobacter*, and *Vogesella.* The dominant flora was *Limnohabitans*, whose abundances in the three samples were 13.17, 14.65, and 17.51%, respectively. The most abundant species in the Jiangbei area were *Limnohabitans*, *unclassified-f-Comamonadaceae*, and the *CL500-29-marine-group*. The dominant flora was *Limnohabitans*, whose abundances in the three samples were 11.18, 11.56, and 11.81%, respectively.

#### 3.3.3. Heatmap Analysis of Community Composition

The heatmap displayed the species composition information of the community by reflecting the number of species through a color ladder. As can be seen in Figure 5, the community composition of the three samples collected from the Jiangbei District and the Beibei District was relatively uniform; one of the three samples collected from the Hechuan District and the Shapingba District had a few differences. The community composition of Hechuan and Beibei was the most similar. *Limnohabitans* [35], *comamonadaceae*, *hgcl_clade*, and *CL500-29-marine-group* had a higher abundance in the four samples; in the Shapingba Distric, it also contained high levels of *Acinetobacter* and *Vogesella*, and Hechuan District also contains high levels of *Aquabacterium*. Bacilli are the predominant flora in urban river water and sediments, which are associated with the gut flora of many mammals and humans, and are effective fecal surrogate indicators. The higher abundance of bacilli in the downstream may be due to the high concentrations of nitrate, ammonia, feces, etc. in medical waste liquid and domestic wastewater [36]. The results of this study showed that the concentration of pollutants gradually increased along the flow direction of the river due to the discharge of domestic and industrial wastewater, which seriously affected the abundance changes of the bacterial community in the river.

#### 3.3.4. Pie Chart

We statistically analyzed the species abundance of the samples at a certain taxonomic level and present the analysis results in the form of pie charts to present the composition of the species in different groups.

Figure 6a reflects the species composition of the Beibei District, which can clearly reflect that *unclassified_f_Comamonadaceae*, *Hgcl_clade*, and the *CL500-29 marine group* were the dominant species in the Beibei District with a relatively high relative abundance that accounted for the total community of 28.8%. Figure 6b reflects the species composition of the Shapingba District, which clearly reflects that *Limnohabitans*, *Vogesella*, and *Acinetobacter* were the dominant species in the Shapingba District. The relative abundance of *Limnohabitans* was relatively high, accounting for 15.12% of the total community. Figure 6c reflects the species composition of the Jiangbei District, which clearly reflects that *Limnohabitans*, *unclassified_f_Comamonadaceae*, and the *CL500-29 marine group* were the dominant species in the Jiangbei District. The relative abundance of *Limnohabitans* was high, accounting for 11.52% of the total community. Figure 6d reflects the species composition of Hechuan District, which clearly reflect that *unclassified_f_Comamonadaceae*, the *Hgcl_clade*, and the *CL500-29 marine group* were the dominant species in the Hechuan District. The relative abundance of *unclassified_f_Comamonadaceae* was high, accounting for 13.30% of the total community. Combining Venn diagrams, community Bar diagrams, Heatmap diagrams of community composition, and pie diagram test results can help us explore the species composition of the communities at the four sampling points, including the number of species and species abundance, by comparing samples or groups. The differences in species composition can reflect the composition information between them more clearly, and provide detailed parameters for monitoring and judging the pollution of the entire Jialing River.

### 3.4. Analysis of Species Differences

#### 3.4.1. Kruskal–Wallis H Test Analysis

In this study, the Kruskal–Wallis H test was used to analyze the species difference of the samples. As seen in Figure 7, each group contained the genus *Limnohabitans*, the genus *unclassified_f_Comamonadaceae,* and the genus of the Hgcl_clade with a relatively high relative abundance. Compared with other groups, the species with significant differences in the Shapingba District were *Acinetobacter* [37], *Vogesella*, and *Pseudomonas*, all of which had a higher relative abundance, whereas *Cavicella* had a lower relative abundance. The species with significant differences in the Hechuan District were *Aquabacterium*, *Novosphingobium*, and *Cavicella* with a higher relative abundance, whereas *norank_Vicinamibacteraceae* had a lower relative abundance. The species with significant differences in the Beibei District had a higher relative abundance of *norank_Vicinamibacteraceae* and a lower relative abundance of *Vogesella.* The species with significant differences in the Jiangbei District were *Acinetobacter*, *Vogesella,* and *Aquabaterium*; all of which had relatively low relative abundances.

#### 3.4.2. LefSe Analysis

A LefSe analysis is used to analyze species with significant differences between samples or groups, reflecting each taxonomic level from high to low from inside to outside. As seen in Figure 8, among the four groups, the species with significant difference in the Beibei District is *Actinobacteriota* and its LDA value is 4.94825; the species with significant difference in the Shapingba District was *Gammaproteobacteria,* which had the greatest impact, and its LAD value was 4.857, The species with a significant difference in the Hechuan District was *Comamonadaceae*, whose LDA value was 4.857 and the species with a significant difference in the Jiangbei District was *Pseudomonadale*, whose LDA value is 4.975. In an ecosystem, there are always high-abundance and low-abundance species. By using Kruskal–Wallis H test analysis and LefSe analysis, it can help us find out the differences between the species at these four sampling points, and successfully construct the ASV set to obtain the differences in the species distribution of different microbial communities, and then to infer the species status of the entire Jialing River in Chongqing. To sum up, hospital waste liquid, agricultural sewage, and urban wastewater contain some antibacterial agents, drugs, and high-concentration pollutants. By exerting selective pressure, these substances promote the emergence of indigenous bacterial diversity, which in turn differentiates high-abundance bacteria in the watershed [38,39]. Therefore, as the main source of drinking water in cities, the discharge of sewage directly poses a threat to human health and the ecological security of the river basin.

### 3.5. RDA/CCA Analysis

Using the RDA map to analyze the relationship between the water environmental factors and the bacterial colonies, we could detect the relationship between the environment, the colony, and the sample as well as their inter-relationships. Studies have shown that pH, turbidity, and nutrients are the dominant factors leading to changes in bacterial community composition. Turbidity is an important driving force for biological metabolism in rivers, and has the greatest contribution to the changes of bacterial community composition in the river basin. Phosphorus and inorganic nitrogen concentrations are generally considered to be important drivers of biological metabolism in flowing water, and phosphorus was shown to be the most determinant factor significantly associated with changes in plankton bacterial communities [40,41]. As seen in Figure 9, ammonia nitrogen, the chemical oxygen demand, and total nitrogen all had significant effects on the distribution of bacterial colonies; among them, ammonia nitrogen had the greatest impact. Ammonia nitrogen had a greater impact on samples 1 and 2 and samples 3 and 2 in the Hechuan District. The influence of chlorophyll A on sample 1 and sample 3 as well as sample 2 and sample 3 in the Shapingba Distric was quite different; samples 1 and 2 and samples 1 and 3 in the Jiangbei District were significantly affected by the environmental factor of chlorophyll A. The figure shows the correlation between the five species and environmental factors. Among them, *Acinetobacter* was the most abundant and its distribution was most affected by chemical oxygen demand. Its colony distribution was positively correlated with chemical oxygen demand, and the ammonia nitrogen, total phosphorus, and chlorophyll A were negatively correlated. The relative abundance of *unclassified_f_Comamonadaceae* was the smallest, and its distribution was most affected by the chemical oxygen demand. The distribution of the colony was positively correlated with ammonia nitrogen and total phosphorus content; it was negatively correlated with chlorophyll A and the chemical oxygen demand.

### 3.6. Evolutionary Analysis

An evolutionary analysis was comprehensively considered using the maximum likelihood method [42]. As seen in Figure 10, the evolutionary distance between *Methylotenera* and *norank_Methylophilaceae* was relatively close. The evolutionary distance between *unclassified_Rhodobacteraceae* and *Pseudorhodobacter* as well as *Novosphingobium* and *unclassified_Sphingomonadaceae* was also relatively close. *Dinghuibacter* and *Sediminibacterium* were relatively close; the *Hgcl_clade* and *norank_Sporichthyaceae* were also relatively close. By an ML maximum likelihood analysis, it was ascertained that the evolutionary distance between *Methylotenera* and *norank_Methylophilaceae* was relatively close. There was an unclassified evolutionary distance between *Rhodobacteraceae* and *Pseudorhodobacter*. *Novosphingobium* had a close evolutionary distance to *unclassified_Sphingomonadaceae*. *Dinghuibacter* and *Sediminibacterium* had a close evolutionary distance and the evolutionary distance between the *Hgcl_clade* and *norank_Sporichthyaceae* was also close. Based on the above experimental results, the analysis of Heatmap and RDA shows that Limnohabitans, unclass_f_comamonadaceae and hgcl_clade are the dominant bacterial groups in the Jialing River waters in Chongqing during the wet season, indicating that the prokaryotic community in this watershed has high genetic diversity. Its distribution is greatly affected by ammonia nitrogen and chemical oxygen demand.

## 4. Conclusions

In this paper, high-throughput sequencing was used to analyze the V3-V4 of 16srDNA, and the microbial community in the Jialing River waters in Chongqing was analyzed from the prokaryotic community diversity, community composition and species differences of the four samples, as well as the correlation analysis with environmental factors. The diversity index showed that there were obvious differences in the biodiversity of the four regions, with the highest biodiversity in Jiangbei District and the lowest in Shapingba. At the genus level, *Limnohabitans*, *unclass_f_comamonadaceae*, and *hgcl_clade* were the main dominant flora with greater abundance and uniformity. The Kruskal–Wallis H test was used to analyze the differences in species composition among the communities. It was found that the Shapingba area has higher abundance of *Limnohabitans*, the Hechuan area has a wide range of *unclass_f_comamonadaceae*, and the Beibei area has higher *hgcl_clade*. By measuring the physical and chemical indicators of the water body, namely total nitrogen, total phosphorus, chemical oxygen demand, and chlorophyll, and using the RDA chart to analyze, the results show that the distribution of microbial colonies is greatly affected by environmental factors in the water body, and the chemical oxygen demand and ammonia nitrogen have a greater impact on the distribution of colonies, and different biological colonies are affected by different environmental factors differently. This subject provides technical parameters for inferring the water quality of the Jialing River, and provides an important basis for the basic theoretical research and environmental monitoring of lake microbial ecology in China. In the following research work, a long-term mechanism should be established for the monitoring of the watershed ecosystem. We should consider sampling in different seasons and years in terms of time, compare and analyze the interannual differences in microbial diversity and structure, and explore their laws, so as to provide a scientific theoretical basis for river water quality monitoring and water ecological changes.

## Figures and Tables

**Figure 1 ijerph-19-05525-f001:**
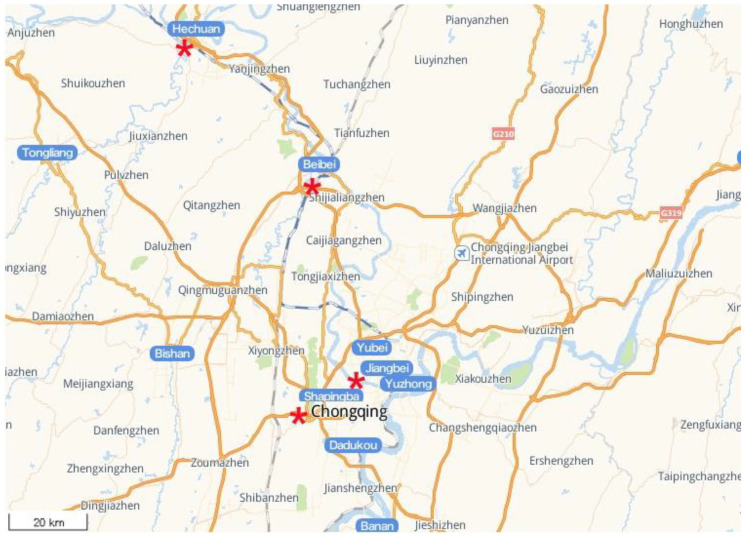
Sample collection site, 1:100,000.

**Figure 2 ijerph-19-05525-f002:**
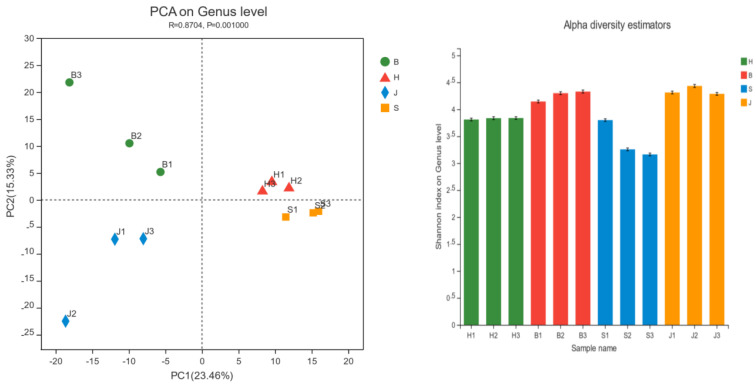
Alpha Shannon index diagram/PCA analysis diagram. (H represents the Hechuan District, S represents the Shapingba District, J represents the Jiangbei District, and B represents the Beibei District).

**Figure 3 ijerph-19-05525-f003:**
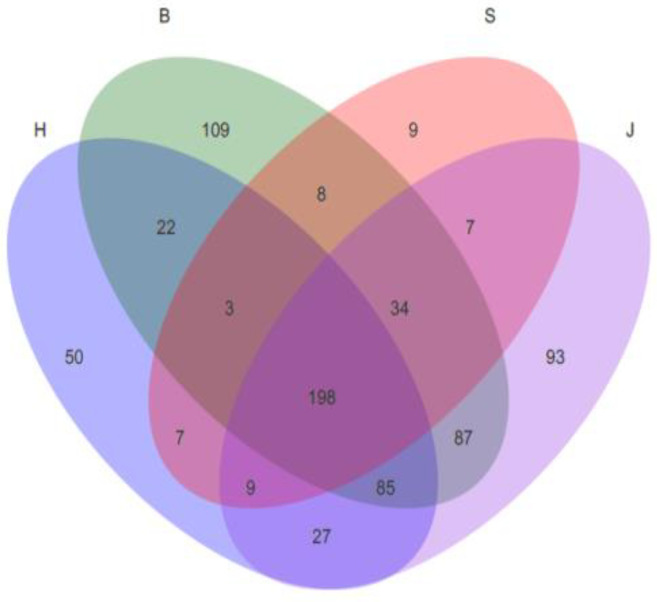
Venn diagram analysis of community distribution.

**Figure 4 ijerph-19-05525-f004:**
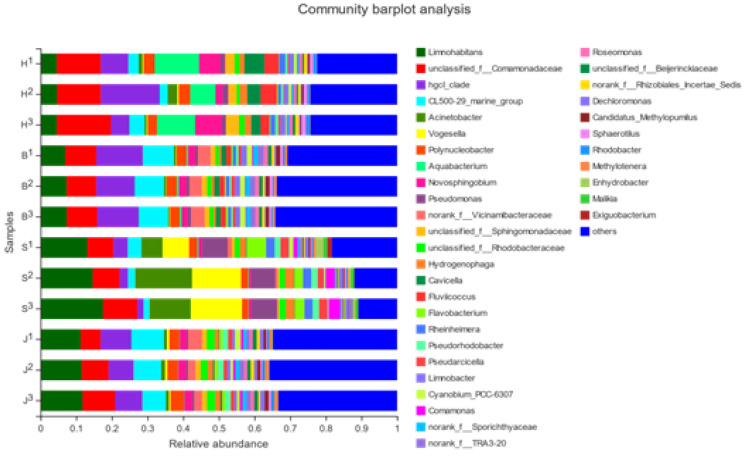
Analysis of the bacterial colony structure at the genus level of the four groups. (H represents the Hechuan District, S represents the Shapingba District, J represents the Jiangbei District, and B represents the Beibei District).

**Figure 5 ijerph-19-05525-f005:**
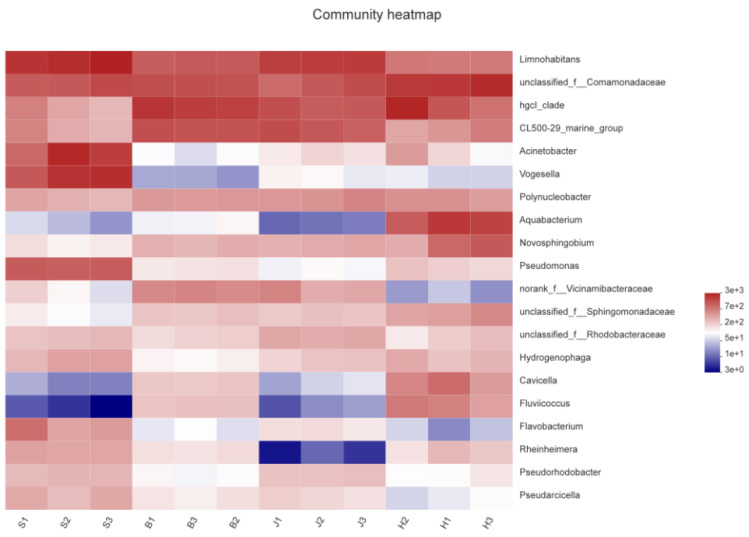
Heatmap analysis.

**Figure 6 ijerph-19-05525-f006:**
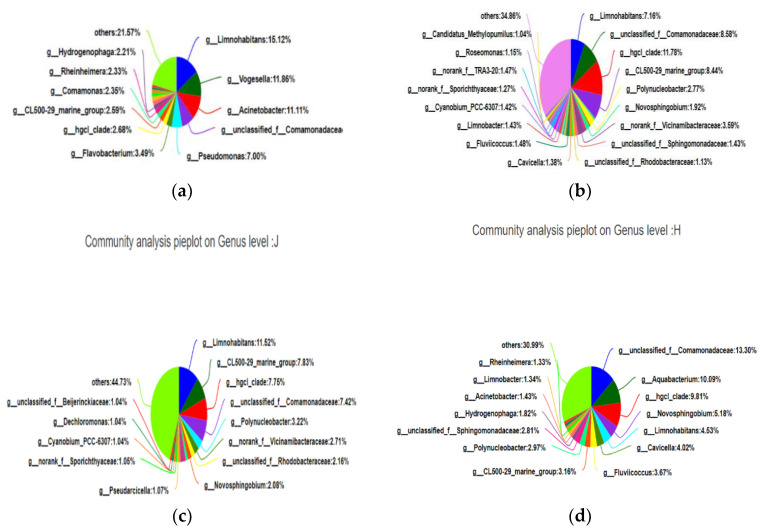
(**a**) Community composition analysis map of the Beibei District, (**b**) community composition analysis of the Shapingba District, (**c**) community composition analysis of the Jiangbei District, and (**d**) community composition analysis of the Hechuan District.

**Figure 7 ijerph-19-05525-f007:**
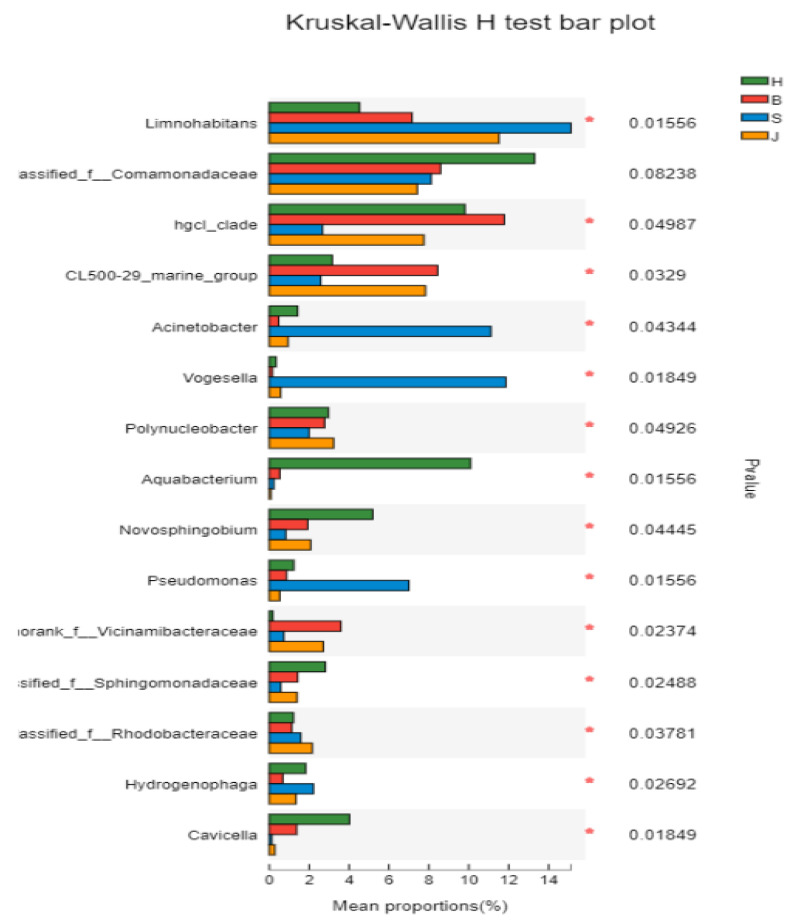
Kruskal–Wallis H test analysis results. (The left side of the vertical axis represents the species name and the right side is the *p*-value. The abscissa represents the relative abundance. H represents the Hechuan District, S represents the Shapingba District, J represents the Jiangbei District, and B represents the Beibei District).

**Figure 8 ijerph-19-05525-f008:**
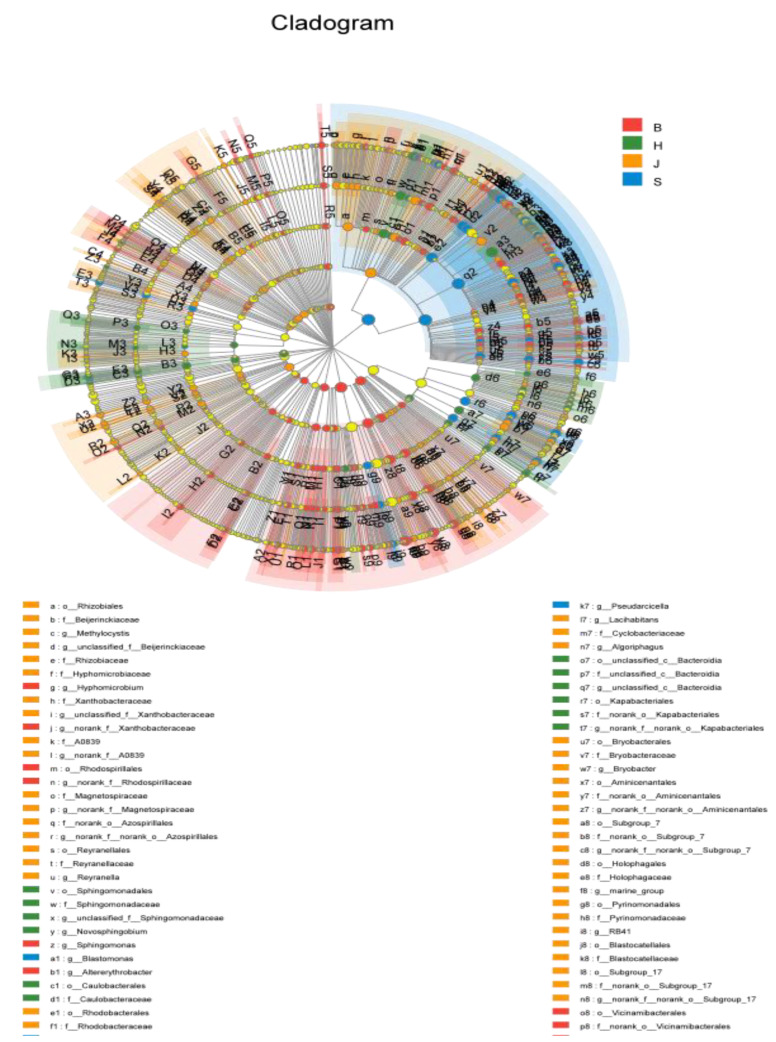
H represents the Hechuan District, S represents the Shapingba District, J represents the Jiangbei District, and B represents the Beibei District; the yellow nodes represent five significantly different species, the colored nodes represent species with significant differences, and the corresponding colors represent the corresponding regions.

**Figure 9 ijerph-19-05525-f009:**
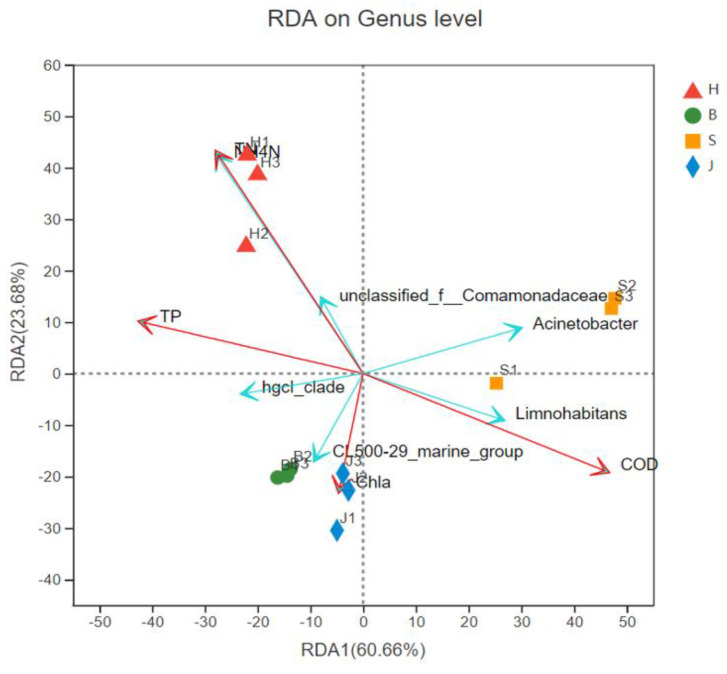
Correlation analysis between community structure and environmental factors. (H represents the Hechuan District, S represents the Shapingba District, J represents the Jiangbei District, and B represents the Beibei District. The longer the red arrow, the greater its influence and the longer the blue arrow, the greater its abundance, COD represents chemical oxygen demand, NH_4_N represents ammonia nitrogen, and Chla represents chlorophyll A.).

**Figure 10 ijerph-19-05525-f010:**
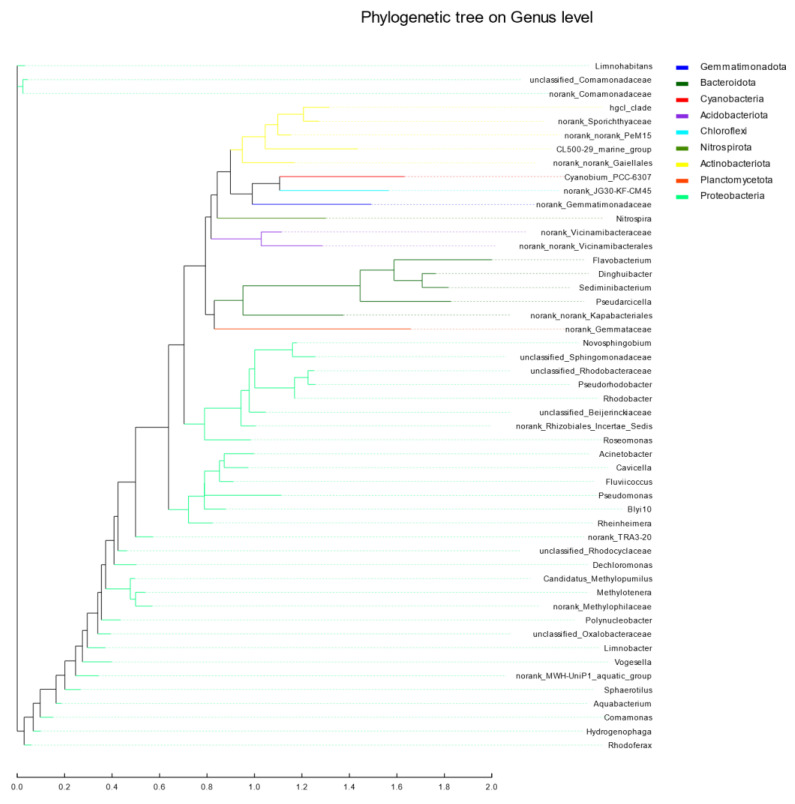
Phylogenetic tree analysis on genus level. (Each branch represents a species and the length of the branch represents the evolutionary distance.).

**Table 1 ijerph-19-05525-t001:** DNA extraction kit product content.

Product Composition	KG203-02 (50 Preps)	KG203-03 (200 Preps)
Buffer B1	6 mL	24 mL
Buffer B2	6 mL	24 mL
2 × Det PCR MasterMix	500 μL	2 × 1 mL
Grinding Pesties	10 ↑	20 ↑

**Table 2 ijerph-19-05525-t002:** PCR reaction system.

Product Composition	Volume
5 × TransStart FastPfu Buffer	4 μL
2.5 mL dNTPs	2 μL
upstream primer (5 uL)	0.8 μL
downstream primer (5 uL)	0.8 μL
TransStart FastPfu DNA polymerase	0.4 μL
Template DNA	10 μL

**Table 3 ijerph-19-05525-t003:** Measurement results of water quality indicators.

Sample Collection Place	Ammonia Nitrogen mg/L	Chemical Oxygen Demand mg/L	Total Phosphorus mg/L	Total Nitrogen mg/L	Chlorophylla μg/L
Shapingba	0.033	24	0.15	0.52	4.1
Hechuan	0.094	7	0.30	1.06	4.2
Jiangbei	0.035	18	0.27	0.51	5.4
Beibei	0.035	12	0.21	0.55	3.9

## Data Availability

The data presented in this article are available on request from the corresponding authors.

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
