# Peer review of "An Analysis of the Colony Structure of Prokaryotes in the Jialing River Waters in Chongqing"

_ijerph, 2022, doi:10.3390/ijerph19095525_

Round 1

Reviewer 1 Report

The summary is written in clear and understandable language, with all the necessary elements.
The material and methods are described in an exhaustive way and allow for an in-depth analysis of the experiment.
Correctly selected research methods, typical for this type of research. the analyzes were conducted very extensively.

Very well presented chapter Results, high level of research results presented.
I recommend it for printing.

Author Response

Dear reviewer,

Thank you for your good comments and suggestions.We have modified the paper as your suggestions,please check it,thank you.

Wish you everything is OK.

Warm regards

Guoming Zeng

Reviewer 2 Report

Authors should reshape the abstract, highlighting the importance of the work in the first sentences and giving also a clear take-home message that reflects the work done. The aim should be revised, for instance, they reported “to explore … harmful microorganisms that cause eutrophication” which were not taken into account throughout the whole study.

Comment to response 1: Despite considerable improvements in the abstract and the introduction, the article still lacks an initial hypothesis and a clear sampling design. Authors should make lines 124 and 125 clearer and better explain the reasons for choosing the four sampling points. The initial hypotheses should be explained in detail. The choice of the environmental parameters could be justified.

The discussion is basically absent. The results are only listed without the reader being guided to their comprehension. In a scientific article, it is necessary to compare the results obtained with previous works for highlighting the results of the research done.

Comment to response 3: many of the concerns I expressed in the first review remain: the discussion and the conclusions are still lacking. In this manuscript, the conclusions are only a summary of the results. In a scientific article discussion and conclusions must be present. Please improve these two paragraphs accordingly.

In addition, I have minor concerns:

The resolutions of all the figures are very low, making it difficult for the reader to understand. Moreover, Figure 1 should be revised and improved in order to make it more comprehensible. Figure 2 is of no use to the study.

Materials and methods should be revised by adding a paragraph for statistical analysis. Please add the references about the primers and the extraction kit used.

Author Response

(The authors gave the same response as above.)

Reviewer 3 Report

Human activities have a significant impact on the quality of water resources. In this work, the microbial community profile of river water and riverbed sediment was studied. The Jialing River in Chongqing was studied at 4 stations for the diversity and composition of the prokaryotic community correlated with environmental parameters. The results showed significant differences in species diversity among the four regions related to water characteristics.

The paper makes an important contribution to the analysis of the microbial community of river water.

I recommend that the authors describe in more detail the physicochemical methods used to study the water.

Also, I think it would be good to emphasize more the contribution of the presented data in practice, i.e., the applicability of the new findings.

Author Response

(The authors gave the same response as above.)

Round 2

Reviewer 2 Report

Thank you for addressing my comments. However, Discussion is still lacking:

Point 3: … The discussion is basically absent. The results are only listed without the reader being guided to their comprehension. In a scientific article, it is necessary to compare the results obtained with previous works for highlighting the results of the research done.

Comment to response 3: many of the concerns I expressed in the first review remain: the discussion and the conclusions are still lacking. In this manuscript, the conclusions are only a summary of the results. In a scientific article discussion and conclusions must be present. Please improve these two paragraphs accordingly.

Response 3: Thank you for your good comments and suggestions. This paper is the first time to discuss the prokaryotic community structure and its relationship with environmental factors in the Jialing River Basin of Chongqing, so there is no previous data for comparison.We have modified the results, discussion and the conclusions, and according to the submission format, we put the results and conclusions together, and the conclusions are presented separately, please check it, thank you.

In this part, the Authors should compare their results with available literature. I have not found this information in the MS. Please, provide a good and robust discussion.

-maps need a scale bar

-Materials and methods in some parts are too detailed. These details should be moved in the supplementary materials and should be concisely described in the main text.

Author Response

Dear rewiewer,

Thank you for your good comments and suggestions. We have modified the maps,Materials and Methods,results, discussion and the conclusions, please check it, thank you.

Warm regards

Guoming Zeng

This manuscript is a resubmission of an earlier submission. The following is a list of the peer review reports and author responses from that submission.

Round 1

Reviewer 1 Report

Add 2 sentences with results to the abstract,

Expand the introduction, add more examples of citations from the literature to support the right choice of the topic of the article,

In the material and methods chapter, complete and add a subsection and describe the applied / used methods and statistical analyzes.

The distribution of results and discussion should generally be separate. But in the current form, there are far too few citations of the literature and there are too few comparisons of the results of the authors' research with other literature reports. I would like to supplement and extend it with the latest research in this field. The presentation of the research results at a high level should be underlined.

Please expand the conclusions as well.

Reviewer 2 Report

Comments and Suggestions for Authors

I felt like I’m reviewing an early draft of my student and couldn’t finish reading through the manuscript. Authors shouldn’t use the review system for their personal use. The entire manuscript needs thorough review and preparation from the scratch including a description of the method (why do you need figure 1?) and presentation of results, to be published as a research article.

Seek a professional English edit service after the contents are ready. The current version has so much help made it very difficult to read and understand what some sentences were trying to convey.

Specific comments

Title: Throughout the manuscript authors were talking mostly about bacteria and sometimes used microorganisms, but rarely used prokaryotes. Just change the title with bacteria. Also “colony structure” confuses readers with the bacterial colonies.

L22: Why two groups “bacteria and harmful microorganisms”? They seem to be odd pairs to be stated here since all bacteria are microbes.

L23. Change to “16S rRNA gene”.

L24-25. Change to “analyzed for the diversity and composition of prokaryotic community, and”

L27. Use more informative full info on those taxa instead of direct copy and paste from the database.

L26-30: Rewrite this part into 2 maybe 3 sentences.

L39-40: I don’t understand this sentence. What do you mean by “becoming more and more close”?

L40-41: I can’t agree with this sentence. You should remove this one or update maybe more geographically specific on Chongqing, if that is the case.

L57-59: I appreciate the authors’ appreciation of microbial ecosystem services but some of them seem overly stretched e.g., flood regulation?

L65-75: This summary of the method is unnecessary. Instead, provide the overall objective of the study.

Throughout the manuscript, I found numerous places that need extensive English editing. The current version is almost prevented me from carefully examining the contents of the manuscript. I’ll try to ignore English edit problems from now on.

2.1. Remove this section and incorporate info where they are used.

Figure 11. Why the first figure is called figure 11? Since the manuscript is not intended for publishing in a Chinese journal, the map needs to be replaced with an English version.

L97-99: What is stored? Filtered water or filter?

L99-100: What do you mean by “16S rDNA transcriptome sequencing”? What was sent to the sequencing lab? Water sample, filter, amplicon…?

I had to stop here since I feel like I’m reviewing an early draft of my student. Authors shouldn’t use the review system for their personal use. The entire manuscript needs a thorough review and work from the scratch including a description of the method (why do you need figure 1?) and presentation of results, to be published as a research article.

Reviewer 3 Report

Summary:

In this study, the community composition of microorganisms in four sampling stations of the Jialing River waters in Chongqing and their potential correlation with environmental factors have been investigated. More in detail, the Authors highlighted diverse bacterial communities through 16s rDNA metabarcoding, which might be shaped by different physico-chemical parameters such as total nitrogen and chemical oxygen demand. The Authors revealed a high occurrence of distinct taxa such as Limnohabitans, unclass_f_comamonadaceae and hgcl_clade in the four sampling stations and significant differences in the prokaryotic community composition between the four groups.

Overall assessment:

I think that the study of the microbial community composition in river waters provides additional information for the overall scientific community.

However, the manuscript is severely lacking the elaboration of data collected which should be provided by the Authors for drawing strong and reliable discussion and conclusions.

In the present work, it is not clearly explained the sampling design and some methodologies which lack fundamental information for reproducing data.

The aim of the present work is not completely clear and should be carefully revised. The Authors should carefully reconsider the figures provided in terms of resolution and information to give to the reader. Moreover, the discussion must be included and the conclusion must be improved based on the scientific results obtained and should rely on the aims of the work.  

In addition, I have several concerns regarding the present study:

  • Authors should reshape the abstract, highlighting the importance of the work in the first sentences and giving also a clear take-home message that reflects the work done.
  • The aim should be revised, for instance, they reported “to explore … harmful microorganisms that cause eutrophication” which were not taken into account throughout the whole study.
  • The introduction should be more focused on river ecosystems avoiding misleading information on other environments such as lakes.
  • Material and Methods: environmental parameters collected during sampling and the sampling design should be better highlighted. The first paragraph is not necessary and should be removed and integrated within the other paragraphs. The Figure 11 should be revised because it is unclear. Different steps of the analysis should not be expressed as a protocol.
  • Analyses and data presentation of microbial taxa should be performed and discussed as low as possible taxonomical level. Authors should indicate the right taxa names and describe the results on the basis of the figures.
  • The discussion is basically absent. The results are only listed without the reader being guided to their comprehension.   In a scientific article, it is necessary to compare the results obtained with previous works for highlighting the results of the research done.
  • Clear and impressive conclusions on the work carried out should be provided.